# Whole exome data prioritization unveils the hidden weight of Mendelian causes of male infertility. A report from the first Italian cohort

Gioia Quarantani[1], Anna Sorgente[1], Massimo Alfano[2], Giovanni Battista Pipitone[3], Luca Boeri[2,4], Edoardo Pozzi[2], Federico Belladelli[2], Filippo Pederzoli[2], Anna Maria Ferrara[2], Francesco Montorsi[2,5], Anna Moles[6], Paola Carrera[3], Andrea Salonia[2,5], Giorgio Casari[1,5]*

1 Genome-Phenome Relationship Unit, Division of Genetics and Cell Biology, IRCCS Ospedale San Raffaele, Milan, Italy, 2 Division of Experimental Oncology/Unit of Urology, URI, IRCCS Ospedale San Raffaele, Milan, Italy, 3 Genomics for Human Disease Diagnosis Unit and Lab of Clinical Genomics, IRCCS Ospedale San Raffaele, Milan, Italy, 4 Department of Urology, Foundation IRCCS Ca' Granda–Ospedale Maggiore Policlinico, University of Milan, Milan, Italy, 5 Vita-Salute San Raffaele University, Milan, Italy, 6 CNR Institute of Biochemistry and Cell Biology, Rome, Italy

* casari.giorgio@hsr.it

## Abstract

Almost 40% of infertile men cases are classified as idiopathic when tested negative to the current diagnostic routine based on the screening of karyotype, Y chromosome microdeletions and CFTR mutations in men with azoospermia or oligozoospermia. Rare monogenic forms of infertility are not routinely evaluated. In this study we aim to investigate the unknown potential genetic causes in couples with pure male idiopathic infertility by applying variant prioritization to whole exome sequencing (WES) in a cohort of 99 idiopathic Italian patients. The ad-hoc manually curated gene library prioritizes genes already known to be associated with more common and rare syndromic and non-syndromic male infertility forms. Twelve monogenic cases (12.1%) were identified in the whole cohort of patients. Of these, three patients had variants related to mild androgen insensitivity syndrome, two in genes related to hypogonadotropic hypogonadism, and six in genes related to spermatogenic failure, while one patient is mutant in *PKD1*. These results suggest that NGS combined with our manually curated pipeline for variant prioritization and classification can uncover a considerable number of Mendelian causes of infertility even in a small cohort of patients.

## Introduction

Infertility affects about 15% of couples, with up to 50% of cases caused by mixed or pure male factors [1]. According to the EAU Guidelines on Sexual and Reproductive Health, both partners should be evaluated in parallel [1, 2].

The diagnostic work-up of male partners includes a comprehensive medical history, physical examination, semen analysis [3, 4], and a basic hormonal evaluation [1, 2]. Karyotype, AZF microdeletions and *CFTR* mutations screening complete the process according to patients'

**Data Availability Statement:** All relevant data are within the manuscript and its Supporting Information files.

**Funding:** The study has been supported by the URI-Urological Research Institute, IRCCS Ospedale San Raffaele, Milan, Italy in the form of internal free funds devoted to translational research to AS and GQ [DONAZ URI. CdC 10A2261]. The funders had no role in study design, data collection and analysis, decision to publish, or preparation of the manuscript.

**Competing interests:** The authors have declared that no competing interests exist.

clinical features and semen parameters [1, 5]. Nevertheless, approximately 40% of infertile men remains undiagnosed [6].

DNA sperm damage, endocrine system dysfunctions, impairment of the testicular microenvironment [7, 8], epigenetic and genetic abnormalities [9] have been suggested to explain idiopathic infertility and hence NGS approach has shown that rare genetic mutations may eventually explain a percentage of those cases.

Recently, the evaluation of these patients by means of NGS approaches has shown that rare genetic mutations can be associated with idiopathic conditions [10–12].

Currently, 120 genes are at least moderately associated to male infertility, i.e., 36 linked to isolated infertility and 84 genes related to syndromic forms, such as primary ciliary dyskinesia (PCD), disorders of sexual development (DSD) and hypogonadotropic hypogonadism (HH) [13].

Among the isolated infertility related genes, those with the higher association are *AR, AURKC, CFAP251, CFAP43, CFAP44, CFAP65, CFAP69, DNAH1, DNAH17, DPY19L2, FANCM, M1AP, MEI1, PLCZ1, PMFBP1, SPEF2, SUN5, SYCP3, TEX11, TEX15, TTC29, CFTR, ADGRG2, ADAD2, DMRT1, GCNA, MSH4, MSH5, NR5A1, RAD21L1, SHOC1, SPO11, SYCE1, SYCP2, TERB1, TERB2, TEX14, and ZMYND15* [10, 11]. 134 additional genes are related to male infertility with an unconfirmed clinical validity since few cases are reported [11]. OMIM (Online Mendelian Inheritance in Man) [14] reports 79 isolated monogenic spermatogenic failure forms (SPGF, PS258150), 46 types of PCD (PS244400), 27 forms of HH (PS147950) and 12 genes related to DSD (PS400044).

To implement the diagnostic yield of infertile couples associated with pure idiopathic male factors while investigating potential unknown causes, we applied whole exome sequencing (WES) on a homogenous cohort of 99 non-Finnish white-European Italian men presenting at a single Reproductive Medicine center. As spermatogenesis is a complex and strictly regulated developmental pathway [15], with 705 genes involved only in the specific process (http://geneontology.org) and 1968 genes expressed in testis (https://www.proteinatlas.org/humanproteome/tissue/testis), genes prioritization represents an instrumental step for the proper genotype-phenotype association [16]. In order to prioritize variants, we designed a manually curated infertility-specific library by including genes already associated to human syndromic and non-syndromic male infertility forms [16]. Results showed that the implementation of this pipeline was able to uncover a significant number of Mendelian causes of infertility even in a small cohort of patients. A deep understanding of male infertility etiology will improve treatment options and enable natural conception, resulting also in the equalization of the burden that, in the ART (Assisted Reproductive Technology) era, falls mainly on women [17]. Furthermore, infertile men and their family members are at an increased risk of developing various types of tumors, whose underlying mechanisms are not fully understood [18]. Further understanding of the genetic impact on infertility will also help to unravel pathways that connect infertility to cancer.

## Materials and methods

### Patients

A cohort of 99 unrelated-infertile non-Finnish white-European men presenting for primary couple's infertility associated with pure idiopathic male factor was selected from the Urological Research Institute (URI) biobank of IRCCS San Raffaele hospital (Milan) between 2017 and 2022. Idiopathic infertility was assumed when our routinary diagnostic protocol [19] gave negative results. Basically, the standard workup includes: (i) parallel assessment of the fertility status, (ii) a complete medical history, (iii) physical examination and semen analysis, (iv) full

andrological assessment including serum total testosterone and Follicle Stimulating Hormone/Luteinising Hormone, (v) testes volume, assessed through a Prader orchidometer, (vi) standard karyotype for diagnostic purposes, (vii) Y-chromosome microdeletion, (viii) cystic fibrosis transmembrane conductance regulator (CFTR) gene mutations.

Men with idiopathic non-obstructive azoospermia (iNOA) were included in the study when having no spermatozoa because of non-obstructive causes in at least two consecutive semen analyses according to the WHO criteria [4]. Patients with the following clinical features were excluded from the study: (i) testicular factors previously associated with infertility (cryptorchidism; grade II and III varicocele; disturbance of erection/ejaculation); (ii) genetic abnormalities previously associated to azoospermia, thus considering CFTR mutations associated with congenital bilateral absence of the vas deferens such as CFTR F508del, heterozygous CFTR F508del, CFTR 5 T/7 T, CFTR 7 T/7 T, and CFTR poly 7 T/9 T, homo and heterozygosis 1298 A > C for the MTHFR gene; microdeletions on the Y chrosomome such as AZFa/b/c; Klinefelter or Kallman syndromes; (iii) known hypothalamic/pituitary defects; (iv) either pituitary or testicular surgery and/or previous vasectomy; (v) previous tumors, including testicular tumors; (vi) testosterone replacement therapy; and, (vii) any other known reason for genital tract obstruction.

Conversely, inclusion criteria were (i) a clinical diagnosis idiopathic infertility associated with primary couple's infertility; (ii) age ≤ 45 years; (iii) white-Caucasian ethnicity; (iv) freedom from any known viral and bacterial infections and antibiotic therapies at the time of surgery, when performed, and throughout the preceding 6 months; and (v) a comprehensive blood set of analyses over the 12 months before surgery, when performed. Authors had access to information that could identify individual participants during or after data collection.

## Ethical approval

Data collection followed the principles of the Declaration of Helsinki; all patients signed an informed consent agreeing to supply their own anonymous information and tissue specimens. The study was approved by the Institutional Review Board (Ethical Committee IRCCS Ospedale San Raffaele, Milan, Italy—Prot. URI001-2010, Feb 14,2014—Pazienti Ambulatoriali), and the recent amendment for the protocol for biobanking (Authorization Protocol URI001-2010, further amended on December 16, 2020). All methods were carried out in accordance with the approved guidelines. Written informed consents were collected from all patients.

## DNA extraction, library preparation and whole exome sequencing (WES)

Genomic DNA was extracted from peripheral blood mononuclear cells using the Maxwell® 48 Instrument (Promega,) and the Maxwell® RSC Blood DNA Kit (AS1400). DNA quantification was performed on Qubit® 3.0 Fluorometer (Broad range Kit, Invitrogen, Q32853), the Nanophotometer® P-Class 300 instruments.

DNA was fragmented using SureSelect Enzymatic Fragmentation kit (Agilent). NGS library were prepared using SureSelect XT HS/Low Input Kit with All Exome V7 RNA Oligos (Agilent). Both DNA fragmentation and library preparation were automated on Hamilton MicroLab STAR M technology.

Library concentration and quality were assessed by Qubit® 3.0 Fluorometer and the 2100 Bioanalyzer Instruments, respectively.

WES was performed on Illumina NovaSeq 6000, S2 flow cells, with a mean coverage 114x.

Reads were aligned against GRCh37 reference. Variant calling was executed with germline pipeline of Dynamic Read Analysis for GENomics (DRAGEN, Illumina). Single nucleotide variants (SNVs) were annotated using ANNOVAR 3.1.2 [20].

## INFERT_Lib design

We generated an infertility-specific library of genes already associated to human syndromic and non-syndromic male infertility forms. The library was manually curated by using Online Mendelian Inheritance in Men (OMIM, https://www.omim.org), and the most recent reviews [11]. The searching terms were "male infertility genes", ("genomics" + "male infertility"), ("molecular genetics" + "male infertility"), "azoospermia", "oligospermia", "teratospermia", "oligoasthenoteratospermia", "asthenospermia", "asthenoteratospermia", "hypogonadotropic hypogonadism", "primary ciliary dyskinesia", ("syndromic" + "male infertility") and "disorder of sex development". UniProt (https://www.uniprot.org), GeneCards and (https://www.genecards.org) The Human Protein Atlas (https://www.proteinatlas.org) have been used to look for information about gene functions, associated disease and tissue expression specificity.

## STRING analysis

We performed a STRING [21] analysis using the Multiple Protein by name function selecting as organism "*Homo*". STRING of INFERT_Lib recognizes 279 out of 283 genes, for k-mean clustering we use k = 3. STRING of syndromic genes mapped 133 out 136 genes, and for non-syndromic 155 out of 156 genes. Genes associated to both forms were considered in both analyses. For k-mean clustering of syndromic and non-syndromic k = 2. To describe clusters, we used the 7 items with the lower FDR, resulted from enrichment with of Biological Process of Gene Ontology.

## Variants prioritization and classification

[15] To prioritize variants, we adopted two analysis pipelines: i) screening for biallelic loss-of-function LoF (nonsense, frameshift, splice-site +/- 1–2) regardless the genes; and ii) considering only variants arising in INFERT_Lib (*S1 Fig* and *S1 Table*).

Variants were filtered for coverage (> 15x), population frequency in GnomAD v2.1.1 [22] (MAF ≤ 0.01 in Global population) and type of variants (missense, nonsense, in-frame indel, frameshift, splicing region +/- 8 bp). Only variants in agreement with the expected mode of inheritance were followed up. *PKD1* missense variants were excluded unless they affect splice sites, and variants in HH genes were not considered if patients hormonal profile was not as expected (T < 3 ng/mL, FSH < 8 mUI/mL, LH < 9.4 mUI/mL). Since high coverage has been considered as a sufficient quality indicator [23], Sanger confirmation has not been performed. Our personal data based on internal procedure validation for more than 2000 genetic variants diagnosed through the NGS, first, followed by Sanger sequencing confirmation, revealed an excellent concordance of variant calling (PC, personal communication).

All variants were classified according to ACMG-AMP guidelines [24–27] and a posterior probability of pathogenicity (post-P) was calculated using Bayesian approach [28]. The criteria have been adapted to analyze infertility-associated variants according to a previously published work [10] (see *S1 Table*).

## Results and discussion

We analyzed 99 idiopathic infertile men *(Table 1)* presenting for primary couple's infertility associated with pure male factor of idiopathic origin, according to WHO criteria [3, 4], with no common genetic abnormalities. Fifty-one (51,5%) patients had non-obstructive azoospermia (NOA); 26 (26.3%) oligoasthenoteratozoospermia (OAT); 8 (8.1%) asthenoteratozoospermia (AT); 7 (7.1%) teratozoospermia (TE); 6 (6.1%) oligoteratozoospermia (OTE); and 1 (1.0%) patient had obstructive azoospermia (OA).

**Table 1. Cohort clinical data.**

| PATIENT ID | PATIENT PHENOTYPE | TOTAL TESTOSTERONE (ng/ml) | FSH (mUI/ml) | LH (mUI/mL) | TESTICULAR HISTOLOGY | RIGHT/LEFT TESTICULAR VOLUME (ml) | TESE OUTCOME | CCI AGE ADJUSTED |
|---|---|---|---|---|---|---|---|---|
| OSR1 | NOA | 1.06 | 74 | 28.4 | Leydig cell hyperplasia | 2/2 | 1 | 0 |
| OSR2 | NOA | 4.08 | 21.7 | 8.9 | N/A | 12/12 | 1 | 3 |
| OSR3 | NOA | 2.45 | 25.7 | 13.2 | Sertoli Cell-Only Syndrome | 5/5 | 0 | 2 |
| OSR4 | NOA | 1.16 | 32.4 | 8 | Sertoli Cell-Only Syndrome | 8/8 | 0 | 0 |
| OSR5 | NOA | 2.28 | 7.1 | 2.1 | N/A | 12/10 | 1 | 1 |
| OSR6 | NOA | 5.63 | 17.54 | 4.1 | Sertoli Cell-Only Syndrome | 10/10 | 0 | 0 |
| OSR7 | NOA | 10.72 | 2.76 | 4 | N/A | 15/15 | 1 | 1 |
| OSR8 | NOA | 2.21 | 2.4 | 1.3 | Morgagni Hydatid (no testicular parenchyma) | 25/20 | 0 | 0 |
| OSR9 | NOA | 1.37 | 1.5 | 3.7 | Sertoli Cell-Only Syndrome | 15/15 | 0 | 0 |
| OSR10 | NOA | 2.54 | 4.75 | 6.18 | Sertoli Cell-Only Syndrome | N/A | 0 | 0 |
| OSR11 | NOA | 4.58 | 8.9 | 6.5 | Normal testicular parenchyma | N/A | 1 | 0 |
| OSR12 | NOA | 5.69 | 12 | 8.1 | Normal testicular parenchyma | 8/8 | 1 | 1 |
| OSR13 | NOA | 3.9 | 9.4 | 4.47 | Spermatocitic arrest | 20/20 | 1 | 0 |
| OSR14 | NOA | 3.14 | 18.1 | 4.6 | Leydig cell hyperplasia | 10/10 | 0 | 1 |
| OSR15 | NOA | 2.66 | 11.43 | 6.77 | Sertoli Cell-Only Syndrome | 10/10 | 0 | 0 |
| OSR16 | OAT | 3.28 | 15.8 | 3.6 | Sertoli Cell-Only Syndrome | 8/10 | 1 | 0 |
| OSR17 | NOA | 2.05 | 64 | 19.3 | Sertoli Cell-Only Syndrome | 4/6 | 0 | 1 |
| OSR18 | OA | 6.43 | 2.5 | 4 | N/A | N/A | 1 | 0 |
| OSR19 | NOA | 2.31 | 11.38 | 4.97 | Sertoli Cell-Only Syndrome | 8/8 | 0 | 0 |
| OSR20 | OAT | 5.97 | 16 | 6.6 | Sertoli Cell-Only Syndrome | 10/10 | 0 | 2 |
| OSR21 | NOA | 1.64 | 14.7 | 6.2 | Sertoli Cell-Only Syndrome | 12/12 | 0 | 2 |
| OSR22 | NOA | N/A | N/A | N/A | N/A | N/A | 0 | N/A |
| OSR23 | NOA | N/A | N/A | N/A | N/A | N/A | 0 | N/A |
| OSR24 | NOA | 5.06 | 44.5 | 6 | Normal testicular parenchyma | 10/10 | 1 | 0 |
| OSR25 | NOA | N/A | N/A | N/A | N/A | N/A | 1 | N/A |
| OSR26 | NOA | 8.55 | 11.1 | 2 | N/A | 15/12 | 1 | 1 |
| OSR27 | NOA | 4.27 | 26 | 11.6 | Sertoli Cell-Only Syndrome | N/A | 0 | 3 |
| OSR28 | NOA | 6.39 | 14.9 | 9.3 | Normal testicular parenchyma | 20/20 | 1 | 0 |
| OSR29 | NOA | 1.89 | 26.4 | 13.2 | Sertoli Cell-Only Syndrome | 8/6 | 0 | 1 |
| OSR30 | NOA | 4.1 | 26.1 | 10.7 | Sertoli Cell-Only Syndrome | 10/12 | 0 | 0 |

*(Continued)*

**Table 1.** (Continued)

| PATIENT ID | PATIENT PHENOTYPE | TOTAL TESTOSTERONE (ng/ml) | FSH (mUI/ml) | LH (mUI/mL) | TESTICULAR HISTOLOGY | RIGHT/LEFT TESTICULAR VOLUME (ml) | TESE OUTCOME | CCI AGE ADJUSTED |
|---|---|---|---|---|---|---|---|---|
| OSR31 | NOA | 3.3 | 22 | 9.5 | Normal testicular parenchyma | 12/12 | 0 | 0 |
| OSR32 | NOA | 4.19 | 6.6 | 2.6 | Complete testicular maturation arrest | 25/25 | 1 | 0 |
| OSR33 | NOA | 6.03 | 5.44 | 11.1 | N/A | N/A | 1 | N/A |
| OSR34 | NOA | N/A | N/A | N/A | N/A | N/A | 1 | N/A |
| OSR35 | NOA | N/A | N/A | N/A | Sertoli Cell-Only Syndrome | N/A | 1 | 0 |
| OSR36 | NOA | 3.2 | 30.1 | 12 | Sertoli Cell-Only Syndrome | N/A | 0 | 0 |
| OSR37 | NOA | 4.23 | 32.8 | 11.5 | Sertoli Cell-Only Syndrome | 6/4 | 0 | 0 |
| OSR38 | NOA | 5.89 | 15.32 | 5.79 | Sertoli Cell-Only Syndrome | 8/8 | 0 | 3, renal carcinoma |
| OSR39 | NOA | 3.75 | 9.9 | 12.9 | N/A | N/A | N/A | N/A |
| OSR40 | NOA | N/A | N/A | N/A | N/A | N/A | 1 | N/A |
| OSR41 | NOA | 5.02 | 15.9 | 11.5 | Normal testicular parenchyma | 10/12 | 1 | 2 |
| OSR42 | NOA | 4.06 | 27 | 7 | Sertoli Cell-Only Syndrome | N/A | 0 | 0 |
| OSR43 | NOA | 2.99 | 26.8 | 12.4 | Sertoli Cell-Only Syndrome | 8/6 | 0 | 1 |
| OSR44 | NOA | 6.79 | 2.7 | 3.8 | N/A | 20/20 | 1 | 0 |
| OSR45 | NOA | 4.69 | 29.8 | 5 | Sertoli Cell-Only Syndrome | 15/10 | 0 | 0 |
| OSR46 | NOA | 6.2 | 14.9 | 14.1 | Sertoli Cell-Only Syndrome | 10/10 | 1 | 0 |
| OSR47 | NOA | 3.24 | 73.9 | 39 | Leydig cell hyperplasia | N/A | 0 | 0 |
| OSR48 | NOA | 5.72 | 10.3 | 6.9 | Sertoli Cell-Only Syndrome | N/A | 0 | 0 |
| OSR49 | NOA | 1.76 | 16.1 | 2.5 | Sertoli Cell-Only Syndrome | 12/10 | 0 | 0 |
| OSR50 | NOA | 5.95 | 12.6 | 9.2 | Sertoli Cell-Only Syndrome | 10/12 | 0 | 0 |
| OSR51 | NOA | 4.93 | 2.8 | 2.83 | Normal testicular parenchyma | 12/15 | 1 | 0 |
| OSR52 | NOA | 6.37 | 14.3 | 5.9 | Normal testicular parenchyma | 15/20 | 1 | 0 |
| OSR53 | AT | 2.01 | 4.6 | 3.1 | N/A | 20/20 | N/A | 2 |
| OSR54 | NOA | N/A | 7.5 | 4.4 | N/A | N/A | N/A | N/A |
| OSR55 | OAT | 5.06 | 3.9 | 2.7 | N/A | 25/25 | N/A | 0 |
| OSR56 | TE | 3.25 | 4.3 | 3.2 | N/A | 15/25 | N/A | 1 |
| OSR57 | AT | 3.57 | 4.2 | 1.4 | N/A | 15/20 | N/A | 1 |
| OSR58 | OAT | 4.05 | 3 | 2.6 | N/A | 20/20 | N/A | 0 |
| OSR59 | AT | 4.67 | 3.1 | 4.6 | N/A | 25/25 | N/A | 0 |
| OSR60 | OAT | 1.47 | 16.9 | 8 | N/A | N/A | N/A | 0 |
| OSR61 | OAT | 6 | 3.5 | 4.7 | N/A | 15/15 | N/A | 0 |
| OSR62 | OTE | 5.97 | 7.9 | 6.7 | N/A | 15/15 | N/A | 0 |
| OSR63 | TE | N/A | N/A | N/A | N/A | 25/15 | N/A | 0 |

(Continued)

**Table 1.** (Continued)

| PATIENT ID | PATIENT PHENOTYPE | TOTAL TESTOSTERONE (ng/ml) | FSH (mUI/ml) | LH (mUI/mL) | TESTICULAR HISTOLOGY | RIGHT/LEFT TESTICULAR VOLUME (ml) | TESE OUTCOME | CCI AGE ADJUSTED |
|---|---|---|---|---|---|---|---|---|
| OSR64 | OAT | 4.59 | 9.5 | 7.2 | N/A | 15/15 | N/A | 0 |
| OSR65 | OAT | 3.97 | 6.1 | 4 | N/A | 10/10 | N/A | 0 |
| OSR66 | OAT | 5.68 | 9.6 | 3.6 | N/A | 20/0 | N/A | 0 |
| OSR67 | OAT | 3.14 | 3.1 | 3.2 | N/A | 25/25 | N/A | 1 |
| OSR68 | AT | 5.66 | 2.6 | 2.5 | N/A | N/A | N/A | 0 |
| OSR69 | TE | 5.8 | 15.5 | 5.2 | N/A | 15/15 | N/A | 0 |
| OSR70 | OAT | 4.66 | 2.9 | 4 | N/A | 25/20 | N/A | 0 |
| OSR71 | OAT | N/A | N/A | N/A | N/A | 15/15 | N/A | 0 |
| OSR72 | OTE | 5.53 | 5.7 | 7.2 | N/A | 15/15 | N/A | 2 |
| OSR73 | AT | 5.88 | 4.5 | 6.3 | N/A | 25/25 | N/A | 0 |
| OSR74 | OTE | 6.11 | 5.8 | 8.3 | N/A | 15/15 | N/A | 0 |
| OSR75 | OTE | 6.11 | 5.8 | 8.3 | N/A | 15/15 | N/A | 0 |
| OSR76 | OTE | N/A | 5.3 | 6.9 | N/A | 15/12 | N/A | 0 |
| OSR77 | OAT | 8.33 | N/A | N/A | N/A | 25/25 | N/A | 0 |
| OSR78 | OTE | N/A | N/A | N/A | N/A | 20/25 | N/A | 0 |
| OSR79 | AT | 3.63 | 1.66 | 1.36 | N/A | 15/15 | N/A | 1 |
| OSR80 | AT | 4.82 | 1.7 | 4.5 | N/A | 25/25 | N/A | 0 |
| OSR81 | TE | 3.6 | 10.76 | 5.05 | N/A | 25/0 | N/A | 2,bilateral testicular tumor |
| OSR82 | OAT | 6.5 | 11 | 5.7 | N/A | 12/12 | N/A | 0 |
| OSR83 | OAT | 5.71 | 25.1 | 9.9 | N/A | 10/10 | N/A | 1 |
| OSR84 | OAT | 4.72 | 2.3 | 3.7 | N/A | 20/20 | N/A | 0 |
| OSR85 | OAT | 8.39 | 17.2 | 7.2 | N/A | 20/20 | N/A | 0 |
| OSR86 | OAT | N/A | N/A | N/A | N/A | 20/25 | N/A | 0 |
| OSR87 | OAT | 4.45 | 2.7 | 3.7 | N/A | 20/20 | N/A | 0 |
| OSR88 | TE | 6.8 | 4.7 | 4.3 | N/A | 25/25 | N/A | 1 |
| OSR89 | OAT | N/A | N/A | N/A | N/A | 12/25 | N/A | 1 |
| OSR90 | OAT | 5.95 | 0.9 | 3.4 | N/A | 20/25 | N/A | 0 |
| OSR91 | TE | N/A | N/A | N/A | N/A | N/A | N/A | 0 |
| OSR92 | AT | 4.58 | 5.1 | 5.8 | N/A | 15/12 | N/A | 0 |
| OSR93 | NOA | 3.4 | 14.4 | 8.7 | N/A | 12/10 | N/A | 0 |
| OSR94 | TE | N/A | N/A | N/A | N/A | 20/20 | N/A | 0 |
| OSR95 | OAT | 4.55 | 5.6 | 3.4 | N/A | 20/20 | N/A | 0 |
| OSR96 | OAT | 5.06 | 4.7 | 6.4 | N/A | 20/15 | N/A | 0 |
| OSR97 | OAT | 4.58 | 7.8 | 3.69 | N/A | 20/20 | N/A | 0 |
| OSR98 | OAT | N/A | N/A | 4.6 | N/A | 25/25 | N/A | 1 |
| OSR99 | OAT | 2.74 | 6.5 | 6.8 | N/A | N/A | N/A | 0 |

All patients have a normal karyotype (46, XY), they are negative to CFTR mutations and Y chromosome microdeletions. Abbreviations: AT: asthenoteratozoospermia, NOA: non-obstructive azoospermia, OA = obstructive azoospermia, OAT: oligoasthenoteratozoospermia, OTE: Oligoteratozoospermia, TE: teratozoospermia, CCI: Charlson Comorbidity Index, 0: negative 1: positive, N/A: not available.

Testicular histology and mTESE outcomes were available for NOA and OA patients. Previous studies [10, 12, 29, 30] identified pathogenic variants with mostly recessive inheritance. Therefore, we first considered clear biallelic LoF mutation, regardless the gene function. Unfortunately, this approach failed to provide relevant results (S2 Table), probably owing to

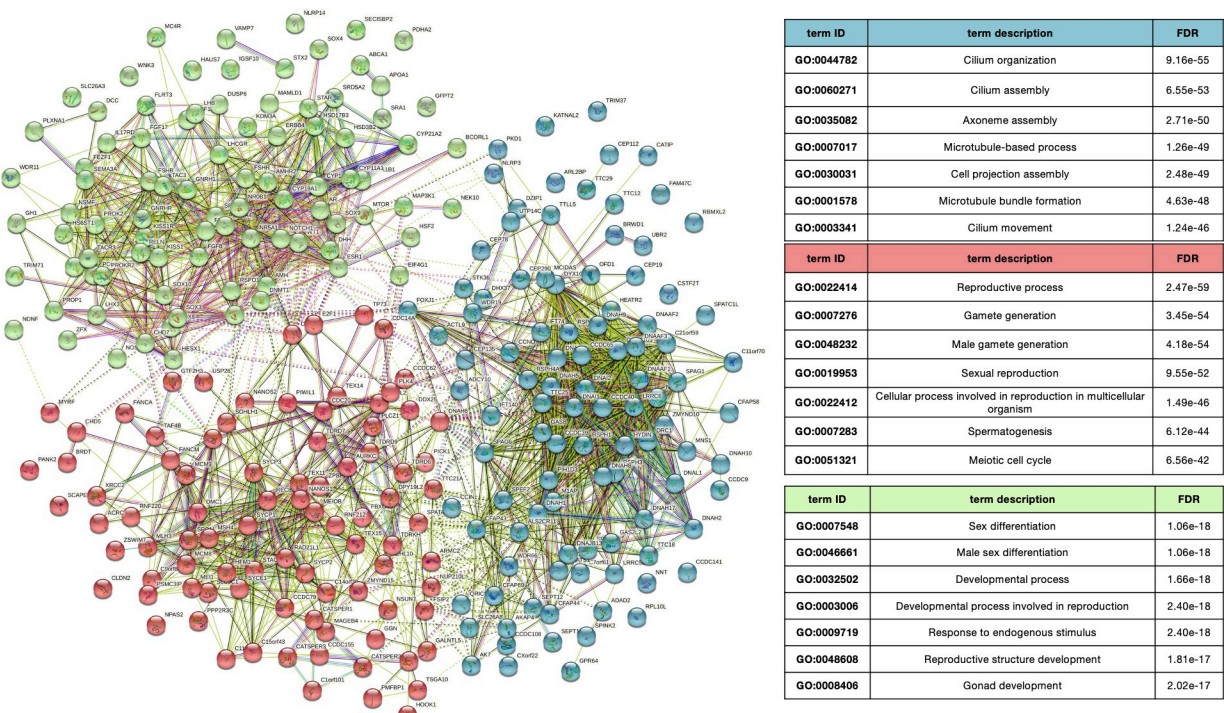

**Fig 1. INFERT_Lib genes forms a highly interconnected network.** STRING analysis highlights an interconnected protein network. analysis of the whole INFERT_Lib (mapped 279 out of 283 genes) with 3 main clusters: genes related to the hypothalamic–pituitary–gonadal (HPG) axis (89, green); genes related to mitosis, meiosis, and cell cycle regulations (89, red); and genes related to flagellum, cilium and acrosome development (101, blue). Several connections between the clusters can also be appreciated (dashed lines).

the composition of our cohort of men with low consanguinity rate. Moreover, it should be considered that several heterozygous LoF variants in genes not related to spermatogenesis were underprioritized.

Subsequently, we considered infertility genes included in the INFERT_Lib, which includes all infertility-associated genes (*S3 Table*). Interactions between part of the selected gene products allowed to group them in three main interconnected clusters (*Fig 1*): i) the hypothalamic–pituitary–gonadal (HPG) axis; ii) mitosis, meiosis, and cell cycle regulation; and iii) flagellum, cilium, and acrosome development. STRING analysis of only non-syndromic genes captures two clusters (*S1A Fig*): i) mitosis, meiosis, and cell cycle regulation and, ii) flagellum, cilium, and acrosome development, which was in common with the only syndromic genes (*S1B Fig*). Conversely, the cluster of genes involved in HPG axis was identified specifically for syndromic genes. The highly interconnected network of genes, already associated to most forms of male infertility, reflects the overall complexity of the spermatogenesis process, which involves the cooperation of several players.

After INFERT_Lib-guided prioritization, we found 189 variants in 62 genes. These variants were then classified according to ACMG guidelines [24] (*S4 Table*). Eleven were *Likely Benign*, 166 *Variant of Unknown Significance* (*VUS*), and 12 *(Likely) Pathogenic*. Overall, causal mutations of monogenic forms of infertility were recognized in 12 patients (12,1%) (3 *Pathogenic* and 6 *Likely Pathogenic* variants; *Table 2*).

Of the seven genes, six are associated to autosomal dominant *(AD)* forms of infertility *(KLHL10, NR5A1, DMRT1, SEPT12, PROKR2, PKD1)*, and one with X-linked recessive inheritance *(AR)*. This highlights that at least in non-Finnish white-European population, there is a

**Table 2. Pathogenic and likely pathogenic variants.**

| Gene | Disease OMIM | Inheritance | MIM number | Patient ID | Patient phenotype | HGVSc | HGVSp | dbSNP ID | Frequency GnomAD ALL | Grantham score | S/PP2/M/C | PhastCons | Domain | ACMG class | TESE outcome |
|---|---|---|---|---|---|---|---|---|---|---|---|---|---|---|---|
| *AR* | Androgen insensitivity syndrome (AIS) | XLR | 313700 | OSR11 OSR21 | NOA | NM_000044.6: c.1174C>T | p.P392S | rs201934623 | 0.004101 | 74 | D/B/D/22,9 | 0.933 | N-terminal domain/Tau5 | LP | 1 / 0 |
| | | | | OSR31 | NOA | NM_000044.5: c.1424C>T | p.A475V | rs200390780 | 0.0015 | 64 | D/B/N/22,8 | 0 | N-terminal domain/Tau5 | LP | 0 |
| *DMRT1* | | AD | 602424 | OSR19 | NOA | NM_021951.3: c.671A>G | p.N224S | rs140506267 | 0.0028 | 46 | D/D/D/32 | 0.998 | | LP | 0 |
| *KLHL10* | Spermatogenic failure 11 (SPGF11) | AD | 608778 | OSR42 OSR85 | NOA OAT | NM_152467.5: c.242A>T | p.N81I | rs36065902 | 0.0006 | 149 | T/P/D/22,6 | 1 | BTB/POZ domain | LP | 0 / N/A |
| | | | | OSR79 | AT | NM_152467.3: c.1038dupG | p.F347Vfs*2 | | | | -/-/D/- | | | P | N/A |
| *NR5A1* | Spermatogenic failure 8 (SPGF8) | AD | 184757 | OSR16 | OAT | NM_004959.5: c.712G>T | p.D238N | rs780568525 | 0.0000764 | 23 | T/B/N/16,95 | 0.984 | Hinge region | LP | 1 |
| *SEPT12* | Spermatogenic failure 10 (SPGF10) | AD | 611562 | OSR59 | AT | NM_144605.5: c.845A>C | p.E282A | rs748928731 | 0.00001768 | 107 | D/D/D/29,9 | 0.998 | GTP binding domain | LP | N/A |
| *PROKR2* | Hypogonadotropic hypogonadism 3 with or without anosmia (HH3) | AD | 607123 | OSR9 | NOA | NM_144773.4: c.254G>T | p.R85H | rs74315418 | 0.0007 | 29 | D/D/D/33 | 1 | 7TM domain | LP | 0 |
| | | | | OSR39 | NOA | NM_144773.4: c.253C>T | p.R85C | rs141090506 | 0.0006 | 180 | D/D/D/32 | 1 | 7TM domain | P | N/A |
| *PKD1* | | AD | 601313 | OSR18 | OA | NM_000296.4: c.9203A>T | p.E3068V | rs1162740312 | 0.000006735 | 121 | T/P/D/22,3 | 0.214 | . | P | 1 |

AR variants are hemizygous, while all other variants are heterozygous. Abbreviations: S: SIFT, PP2: PolyPhen2, M: Mutation Taster; C: CADD phred score, TM: transmembrane P: Pathogenic, LP: Likely Pathogenic.TESE: 0, negative; 1, positive; N/A: not available.

higher prevalence of autosomal dominant and X-linked forms of male infertility. Of these, two patients carry variants in *PROKR2*, which is associated to HH; one patient has a variant in *PKD1*; three patients have variants in *AR*, associated to androgen insensitivity syndrome (AIS). The remaining patients present with variants in SPGF genes such as *KLHL10* (n = 3), *DMRT1* (n = 1), *NR5A1* (n = 1), and *SEPT12* (n = 1).

Both NOA patients OSR11 and OSR21carry a hemizygous missense variant in *AR*, P392S, already reported in [31, 32] and classified as *Likely Pathogenic* according to ACMG criteria (ClinVar: VCV000216890.5). The variant causes a strong amino acid change in the protein sequence (Grantham score, 74) and the residue is strongly conserved across the evolution (PhastCons, 0,933). *In silico* pathogenicity predictors overall assign a deleterious impact on the protein function (Sift, D; PolyPhen 2, B; Mutation Taster, D; CADD, 22,9) *(Table 2)*. All these features support the pathogenic significance of the variant in the context of male infertility.

NOA patient OSR31 displays the missense variant A475V classified as *Likely Pathogenic (Table 2)* in the *AR* gene. In this case as well, the variant effect on the ammino acid change is impacting (Grantham score, 64) but in silico pathogenicity predictors show conflicting results (Sift, D; PolyPhen 2, B; Mutation Taster, N; CADD, 22,8) and the residue is not very well conserved across the evolution (PhastCons, 0,933) *(Table 2)*. Nevertheless, A475V has been reported to reduce the ability of *AR* to activate target promoters *in vitro* [33]. This functional evidence demonstrates indeed the A475V detrimental effect on the AR function.

Both *AR* variants (P392S and A475V) are also reported in the androgen mutation database [34] as associated with partial and mild AIS. They locate in the Tau5 region of N-terminal domain (NTD) *(Fig 2A)*. Tau5 region is directly involved in ligand-dependent interdomain interaction between NTD and the ligand binding domain, which is fundamental to regulate androgen-dependent genes [35]. AR is a transcription factors (TF) that orchestrates the spermatogenesis and it is fundamental for the regulation of each phase [15]. In testes, *AR* is expressed by Leydig and Sertoli cells, namely nurse cells that support differentiating germ cells [15]. Hence, it is not surprising that hypomorphic alleles cause the failure of germ cell differentiation.

NOA patient OSR19 harbors the *Likely Pathogenic* variant N224S in *DMRT1* [36] *(Fig 2B, Table 2)*. Apart from its suggestiveness *(Table 2)*, N224S has been already reported in three infertile patients (Clinvar: VCV000243009.20; 2xVUS infertility associated, 1xB and 1xLB with no associated phenotype) [36]. *DMRT1* is a TF involved in spermatogonia maintenance by inhibiting meiosis and promoting mitosis in undifferentiated spermatogonia [37]. This is consistent with the Sertoli Cell Only Syndrome (SCOS) phenotype of our OSR19 patient, since inactivating mutations in *DMRT1* can lead to depletion of spermatogonia and, in turn, of the other, more differentiated, germ cell populations.

NOA patient OSR42 and OAT patient OSR85 carry a *Likely Pathogenic* missense variant in *KLHL10*.

KLHL10 is involved in ubiquitination and protein degradation [38]. It is composed of a BTB domain, which interacts with CUL3 (component of E3 ubiquitin-ligase complex); a BACK domain, which seems to be involved in substrate orientation; and six kelch-repeats which bind substrate that will be ubiquitinated [38].

*Klhl10* haploinsufficient male mice are infertile due to maturation arrest at late spermatids, and some seminiferous tubules present SCOS phenotype [39]. Hence, Klhl10 is essential to complete spermatogenesis [39].

In humans, KLHL10 is responsible for the autosomal dominant spermatogenic failure-11 (SPGF11; MIM number: 615081) that is characterized by oligozoospermia, and in some patients also by asthenozoospermia and teratozoospermia.

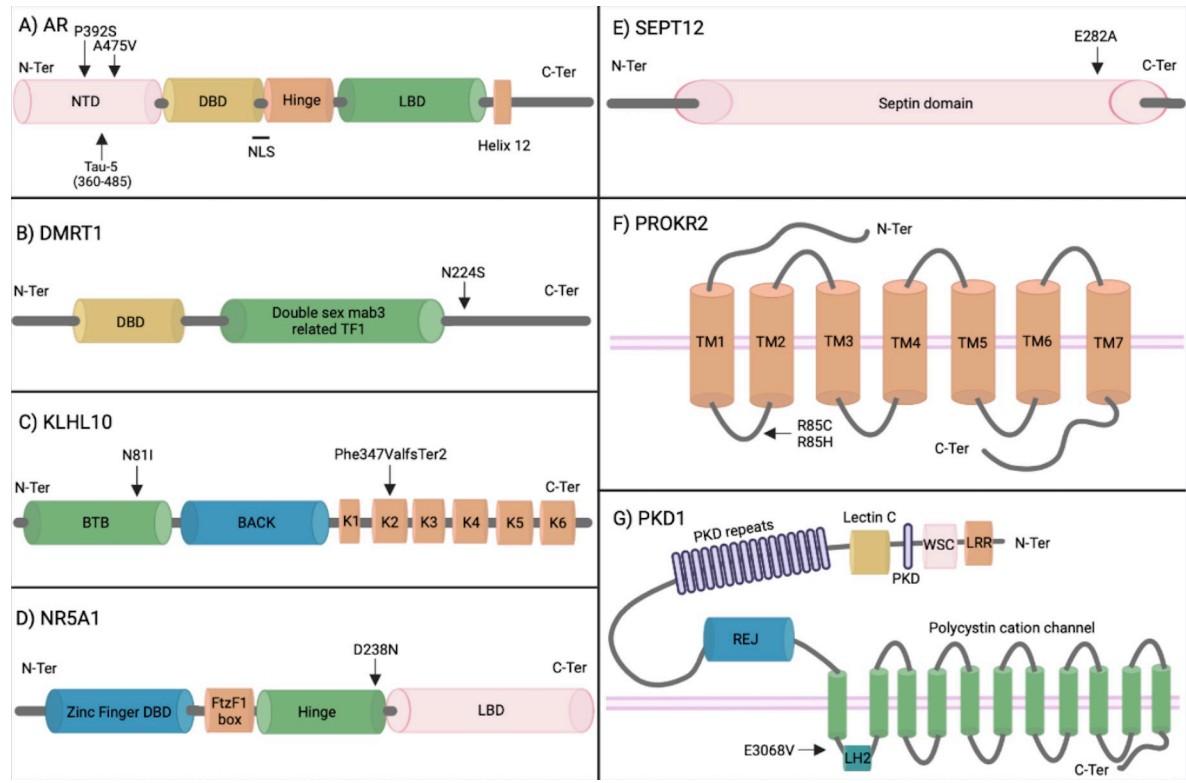

**Fig 2. Protein domain schemes with (Likely) pathogenic variants localization. A)** Functional domain of AR: DBD = DNA binding domain, LBD = ligand binding domain, NLnuclear localization signal, NTD = N-terminal domain. P392S (OSR11, OSR21) and A475V (OSR31) localize in Tau5 region of NTD. Functional domain of DMRT1: DBD = DNA binding domain and double sex mab3 related TF1 domain. N224S (OSR19) is not in any functional domain. **C)** Functional domain of KLHL10: BTB = Broad-Complex, Tramtrack and Bric a brac domain, BACK and K = kelch-repeat. N81I (OSR42, OSR85) localizes in BTB, and F347Vfs*2 (OSR79) in the second kelch repeat. **D)** Functional domain of NR5A1: Zinc finger DBD = DNA binding domain, FtzF1 box, hingeregion, ligand binding domain. D238N (OSR16) localizes in the last amino acid of the hinge region. **E)** SEPT12 has only one functional domain, septin domain, in which E282A (OSR59) localizes. **F)** Domain of PROKR2: TM1-7 = 7 transmembrane domains. R85C (OSR39) and R85H (OSR9) localize in the first internal loop of the 7 transmembrane domains. **G)** Functional domain of PKD1: LRR = leucine rich repeats, WSC domain, Lectin C, PKD repeats, REJ = receptor for egg lelly doamin, polycystin cationic channel, LH2. E3068V (OSR18) localizes in an internal loop of polycystin cationic channel.

Both patients carry the N81I, which localizes in the BTB domain (*Fig 2C*), which is involved in homodimerization and interaction with CUL3, a component of E3 ubiquitin protein-ligase complex [38].

Also in this case, the variant is strongly suggestive of the damaging effect on KLHL10 (Grantham score, 149, Sift, T; PolyPhen 2, P; Mutation Taster, D; CADD, 22,6, PhastCons, 1) (*Table 2*). As a matter of the fact, N81I has been previously reported in 3 patients, one NOA and two oligoasthenozoospermic [40, 41] (ClinVar VCV000684736.2).

The AT patient OSR79 has instead a novel *Pathogenic* frameshift in *KLHL10*, F347Vfs*2, which truncates the protein in the second kelch-repeat out of six [38] (*Fig 2C*). Kelch-repeats bind the substrate that will be ubiquitinated by the KLHL10-CUL3 ubiquitin protein-ligase complex [38], so that a truncated protein cannot bind its substate anymore. This is consistent with the SPGF11 male infertility phenotype.

OAT patient OSR16 carries the *Likely Pathogenic* D238N variant in *NR5A1 (Table 2)*, already been described [42] and classified as *Pathogenic*. D238N localizes in the hinge region of NR5A1[42] (*Fig 2D*) and behaves as hypomorphic as demonstrated *in vitro* [42]. D238N is mostly linked to NOA phenotype and additionally it has been rarely associated to OAT

phenotype [42]. As for *DMRT1* and *AR*, *NR5A1* is a TF fundamental for testes development and spermatogenesis regulation [43]. The Leydig cell *Nr5a1* conditional knockout mouse develops hypoplastic testes in which seminiferous tubules are devoid of lumen and there are no progressing spermatogonia [43]. Our patient has a similar defect characterized by spermatogonia presence in only 5% of his seminiferous tubules, as reported in the clinical record.

AT patient OSR59 carries the *Likely Pathogenic* E282A variant in the GTP binding domain of *SEPT12 (Table 2),* already associated to dominant forms of male infertility (*Fig 2E*). Two infertile patients were described as carriers of heterozygous missense mutations in the GTP-binding domain of *SEPT12* (T89M, D197N) [44]. Functional assays show that both reported variants reduce SEPT12 ability to form Septin oligomers [44]. SEPT12 orchestrates the annulus formation, which is fundamental for proper sperm structure and motility [45]. Patient OSR59 presents both defects, thus making consistent the causal relationship between E282A in SEPT12 and his infertility outcomes.

NOA patients OSR9 and OSR39 carry two different missense variants hitting the same amino acid of *PROKR2 (Table 2)*, R85H (*Likely Pathogenic*) and R85C (*Pathogenic*) respectively. These variants, falling in the first internal loop of the transmembrane domain (*Fig 2F*), are reported in ClinVar as *Pathogenic* or *Likely Pathogenic* for HH and related to patients both with and without anosmia (VCV000003451.25, VCV000156562.7). Both variants behave as loss-of-function *in vitro*, hence pointing out a likely haploinsufficient mechanism [46]. *PROKR2* is associated with an autosomal dominant HH form (MIM: 244200).

However, although both OSR9 and OSR39 patients present pathogenic variants, only patient OSR9 displays a hormonal profile expected for HH (T<3ng/mL, FSH<8 mUI/mL, LH<9.4mUI/mL). Nevertheless, at the clinical evaluation, no etiological factor for his HH phenotype was found, thus he was defined as idiopathic. On the other hand, OSR39, carrier of a HH gene variant, shows high T, FSH and LH *(Table 1)*. This discrepancy between the hormonal profile and the mutation has been already reported in an anecdotical case [47].

Based on ACMG criteria, the missense, rare (<0,0001%), splice site variant E3068V in *PKD1* of OA patient OSR18 is classified as *Pathogenic* (*Fig 2G*). Mutations in PKD1, besides the renal disease, have been associated to defects in sperm morphology, motility, and number [48]. Recently, a frameshift mutation has been reported in an ADPKD male patient suffering also from OA, due to the presence of cysts in both epididymes [49]. Moreover, *pkd1-/-* mice show defects in male reproductive system development, with cystic dilation of the efferent ducts [50]. Therefore, it is likely that E3068V plays a role in the etiology of our patient's OA phenotype.

According to the European Association of Urology guidelines on Sexual and Reproductive Health [51], in addition to the physical examination, the semen analysis, and the hormonal evaluation, infertile male diagnostic work-up includes only karyotype analysis, CFTR mutations and AZF microdeletions screening. However, our high (12%) yield of diagnosis by P/LP variants of genes already associated to infertility, suggests a strong monogenic component in this disorder and highlights the need to implement the diagnostic work-up by adding NGS analysis of a large panel of infertility genes, which could greatly increase the diagnostic effectiveness and possibly open to tailored management of idiopathic infertile patients. A final attribution of the pathogenic burden to the identified gene variants will be achieved by *in vitro* and *in vivo* functional studies and possibly the confirmation on larger cohorts.

In NOA patients, c/mTESE may achieve a positive sperm retrieval for subsequent ART application. Hence, we attempted to associate TESE outcomes to mutant genes as success predictive markers (*Fig 3*). Patients OSR42, OSR19 and OSR9 carrying *KLHL10*, *DMRT1* and *PROKR2* mutations had negative sperm retrieval at TESE. Moreover, two patients (OSR21, OSR31) out of three carrying *AR* variants had a negative outcome. This observation, along

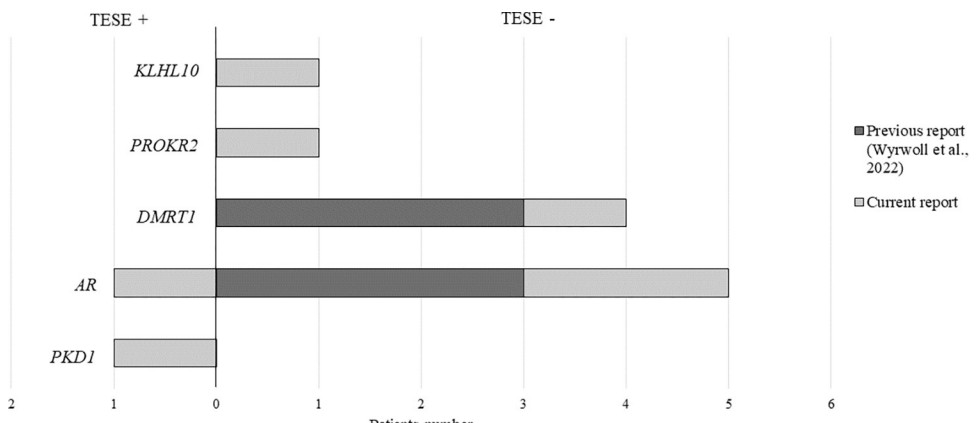

**Fig 3. TESE outcome associated to mutant genes.** Relation between mutant gene and TESE outcome (TESE+ = positive sperm retrieval; TESE— = negative sperm retrieval) in our NOA patients carrying pathogenic variants and in a cohort of a previous report (Wyrwoll *et al.*, 2022).

with previously reported cases [10], increases the evidence of a negative correlation between *DMRT1* mutations and successful sperm retrieval at surgery. Patients with a negative sperm retrieval at TESE are mostly characterized by a SCOS phenotype. As expected, OA patient OSR18 carrying a *PKD1* variant had a positive sperm retrieval at TESE. Therefore, our results suggest that variant of specific changes may represent a relevant predictive biomarker of sperm retrieval, but current findings need to be validated to achieve adequate reliability over the real-life clinical work-up.

In addition, 63 infertile men (62.4%) were carriers of one or more VUS of INFERT_Lib genes. Of all, 28 patients had a single variant, the remaining 35 carried more than one variant (*S4 Table*). Of those latter 35, two patients presented two variants in two genes which may cooperate in terms of infertility etiology (*Table 3*). Indeed, NOA patient OSR8 is carrier of *NOTCH1* (R1661Q) and *ERBB4* (H374Q) (*Table 3*). Given that both genes are players of the HH pathway, we propose a digenic inheritance to be further explored in this context [11]. Furthermore, HH genes have been already associated to digenic inheritance [52]. Variant in *DHX37* is probably neutral because mutations in *DHX37* have been associated to high serum levels of FSH and LH (MIM: 273250), although our patient (OSR8) had low levels of both gonadotropins (i.e., LH = 1.3 mUI/mL; FSH = 2.4 mUI/mL).

In NOA patient OSR50 we observed two missense variants in *DMRT1* (Q302H) and in *NR5A1* (A154T) (*Table 3*). The A154T variant is reported to alter the ability to activate *NR5A1* target promoters [53]. NR5A1 and DMRT1 cooperate in the transcriptional regulation of testis differentiation, both acting on *SOX9* [37]. NOA OSR50 had also a variant in *FAM47C* associated to varicocele patients (*Table 3*) [54].

In all other patients (*S5 Table*) carrying more than one *VUS*, it is arduous to propose the causative variant(s) until future reclassification. Several of these variants may also undergo reclassification as (likely) pathogenic after further evidence (e.g., identification of additional patients with the same variant, functional studies execution, or variant phase determination). This will enable the identification of more causal mutations. Variant phase analysis allows to exclude mutations that are not compatible with the mode of inheritance. Indeed, genetic variants related to both male and female infertility should be *de novo* mutations, appearing for the first time in the infertile patient. Instead, genetic variants linked only to male infertility can be *de novo* or maternally inherited.

**Table 3. Putative digenic forms.**

| Patient ID | Patient phenotype | Gene | HGVSp | HGVSp | dbSNP ID | Frequency GnomAD ALL | Grantham | S/PP2/M/C | PhastCons20 | Domain | ACMG class |
|---|---|---|---|---|---|---|---|---|---|---|---|
| OSR8 | NOA | NOTCH1 | NM_017617.5: c.4982G>A, | p. R1661Q | rs1163223024 | 0.0000234 | 43 | D/D/D/32 | 1 | | VUS |
| | | ERBB4 | NM_005235.3: c.1122T>G | p. H374Q | rs76603692 | 0.002520 | 24 | T/B/D/3,788 | 1 | Leucine-rich repeat domain | VUS |
| | | DHX37 | NM_032656.4: c.2396A>G | p. Y799C | rs147727115 | 0.0002812 | 194 | D/D/D/25,2 | 0.998 | Helicase associated domain | VUS |
| OSR50 | NOA | NR5A1 | NM_004959.5: c.460G>T | p. A154T | rs761496130 | 0.00002188 | 58 | T/B/N/0,055 | 0.205 | | VUS |
| | | DMRT1 | NM_021951.3: c.906G>C | p. Q302H | rs200069202 | 0.00003977 | 24 | D/D/D/28,4 | 0.997 | | VUS |
| | | FAM47C | NM_001013736.3: c.240A>T | p.K80N | rs782218896 | 0.0001 | 94 | T/D/N/11,17 | 0.004 | FAM47 family | VUS |

OSR8 has variant in two HH-related genes, NOTCH1 and ERBB4. OSR50 patients has three variants, those in NR5A1 and DMRT1 have a role in determining NOA phenotype, while those in FAM47C in his varicocele. All variants are in heterozygous state, except for variant in FAM47C that is hemizygous. Abbreviations: S: SIFT, PP2: PolyPhen2, M: Mutation Taster, C: CADD phred score, VUS: Variant of Unknown Significance.

Notably, in several cases we observed mutations in genes that are known oncogenes or onco-suppressors, such as *PLK4* [55] and *CDC20* (53)(52) [56]. Indeed, NOA patient OSR20 carries two heterozygous missenses in *CDC20* (E237G and V361I) and developed osteosarcoma *(Table 1, S5 Table)*. Likewise, TE patient OSR81 has a missense in PLK4 (P317L), and he was diagnosed with testicular cancer *(Table 1, S5 Table)*. The same *PLK4* variant was found in another infertile man (NOA patient OSR4) and in a patient of an in-house cohort of testicular cancer patients *(Table 1, S5 Table)*.

In this context, although not yet proven, it has been postulated that a common genetic factor could account for male infertility etiology and early comorbidity development at least in some cases [57]. Most of INFERT_Lib genes do not have a structural role in the spermatozoon, but they are involved in the cell cycle, mitosis, regulation of transcription and translation. In addition, most of these genes are not exclusively expressed in testes. Since these genes are involved in pathways which alteration can lead to tumorigenesis, it is likely that they are potentially associated both with male infertility and cancer development. Accordingly, some of our patients (OSR20 and OSR81) have germline mutations in genes associated to cancer development–e.g., *CDC20* [56], *PLK4* [55]–that could explain not only their infertility, but also their neoplastic development.

Moreover, one of the most frequently mutated genes in male infertility—*AR*—is itself cancer-associated [35, 58]; in fact, its variant P392S was described in testicular cancer [58]. Although our *AR* mutant patients do not show tumors so far, a follow-up strategy can be envisaged. These cases support the hypothesis that a single mutant gene is involved in both infertility etiology and tumor predisposition. Further investigations are needed to discover the functional link between mechanistic events characterizing these two complex pathways.

Our data are consistent with the evidence that part of idiopathic male infertility cases can be modelled as monogenic diseases with wide genetic heterogeneity and confirm results of a recent study [47] showing that in a large cohort of non-related men, Mendelian causes of NOA infertility are distributed across a vast number of genes involved in testis function, with

most of the variations occurring in singleton cases rather than recurring in different patients. Therefore, sharing experiences for diagnosis and treatment, and to construct case series, will be essential to optimize patient care.

Indeed, a more detailed knowledge of potential monogenic or oligogenic conditions associated with idiopathic male infertility would enable to (i) update the male infertility diagnostic pipeline, by the implementation of a large gene panel sequencing; (ii) better tailor ART strategies (e.g., performing c/mTESE only in those patients with mutant genes positively correlated with successful sperm retrieval); and, (iii) early identify those patients carrying mutations associated with infertility and predisposing to cancer, therefore promoting the implementation of personalized prevention and follow-up strategies, with a relevant rebound on the National Health Systems.

## Supporting information

**S1 Fig. Syndromic and non-syndromic genes form a highly interconnected network.**
(DOCX)

**S1 Table. ACMG criteria description.**
(XLSX)

**S2 Table. Biallelic LoF variant in the whole cohort.**
(XLSX)

**S3 Table. List of INFERT_Lib genes.**
(XLSX)

**S4 Table. ACMG classified variants selected with INFERT_Lib.**
(XLSX)

**S5 Table. Identified VUS variant in INFERT_Lib genes.**
(XLSX)

## Acknowledgments

We greatly acknowledge *ab medica s.p.a*. for instrumentation support.

## Author Contributions

**Conceptualization:** Giorgio Casari.

**Data curation:** Gioia Quarantani, Anna Sorgente, Massimo Alfano, Paola Carrera, Andrea Salonia.

**Formal analysis:** Gioia Quarantani, Anna Sorgente, Anna Moles, Giorgio Casari.

**Funding acquisition:** Andrea Salonia.

**Investigation:** Paola Carrera, Giorgio Casari.

**Methodology:** Gioia Quarantani, Anna Sorgente, Giovanni Battista Pipitone, Luca Boeri, Edoardo Pozzi, Federico Belladelli, Filippo Pederzoli, Anna Maria Ferrara.

**Resources:** Massimo Alfano, Luca Boeri, Edoardo Pozzi, Federico Belladelli, Filippo Pederzoli, Anna Maria Ferrara, Francesco Montorsi.

**Supervision:** Giovanni Battista Pipitone, Anna Moles, Paola Carrera, Giorgio Casari.

**Validation:** Gioia Quarantani.

**Visualization:** Gioia Quarantani, Massimo Alfano.

**Writing – original draft:** Anna Sorgente, Anna Moles, Andrea Salonia, Giorgio Casari.

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
