## [Decision Letter · Decision Letter 0]

5 Jun 2023

PONE-D-23-10435Whole exome data prioritization unveils the hidden weight of Mendelian causes of male infertility. A report from the first Italian cohort.PLOS ONE

Dear Dr. Casari,

Thank you for submitting your manuscript to PLOS ONE. After careful consideration, we feel that it has merit but does not fully meet PLOS ONE’s publication criteria as it currently stands. Therefore, we invite you to submit a revised version of the manuscript that addresses the points raised during the review process.

We look forward to receiving your revised manuscript.

Kind regards,

Nejat Mahdieh

Academic Editor

PLOS ONE

2. Please upload a new copy of Figure 1 as the detail is not clear. Please follow the link for more information: https://blogs.plos.org/plos/2019/06/looking-good-tips-for-creating-your-plos-figures-graphics/" https://blogs.plos.org/plos/2019/06/looking-good-tips-for-creating-your-plos-figures-graphics/

Reviewers' comments:

Reviewer's Responses to Questions

**Comments to the Author**

1. Is the manuscript technically sound, and do the data support the conclusions?

Reviewer #1: Yes

Reviewer #2: Yes

Reviewer #3: Yes

2. Has the statistical analysis been performed appropriately and rigorously? 

Reviewer #1: Yes

Reviewer #2: Yes

Reviewer #3: Yes

3. Have the authors made all data underlying the findings in their manuscript fully available?

Reviewer #1: Yes

Reviewer #2: Yes

Reviewer #3: Yes

4. Is the manuscript presented in an intelligible fashion and written in standard English?

Reviewer #1: Yes

Reviewer #2: Yes

Reviewer #3: Yes

5. Review Comments to the Author

Reviewer #1: This study evaluates the unknown potential genetic causes in couples with pure male idiopathic infertility by applying variant prioritization to whole exome sequencing. This result is very interesting in that it provides a new understanding of male infertility etiology including Mendelian causes of infertility. I think that further understanding of the genetic impact on infertility need to improve treatment options of infertility.

Reviewer #2: The authors exam the gene of idiopathic infertile men. Even though the treatment is same, it's a new field for idiopathic infertile men, and maybe further become routine work up. Several gene is related to cancer in POI women. The author also mention some of the gene is also related to cancer in idiopathic infertile men.

Reviewer #3: The author of the paper entitled "Whole exome data prioritization unveils the hidden weight of Mendelian causes of male infertility. A report from the first Italian cohort" analyzed the whole exome sequencing of the males with idiopathic infertility. I have some comments:

1. only 12 patients of the studied cases diagnosed to have pathogenic and likely pathogenic variants. Please discuss by details (case by case) how you are sure that their infertility is because of the identified variants and also discuss the defect of this method (NGS) to diagnose this type of idiopathic infertility.

2. Insert TESE outcome Column to Table 2 for all 12 patients and discuss the relationship of the identified variants to TESE outcome.

6. PLOS authors have the option to publish the peer review history of their article (what does this mean?). If published, this will include your full peer review and any attached files.

Reviewer #1: **Yes: **Bosun Joo

Reviewer #2: No

Reviewer #3: No

---

## [Author Response · Author response to Decision Letter 0]

12 Jun 2023

Authors’ answers to Reviewers’ comments

5. Review Comments to the Author

Reviewer #1: This study evaluates the unknown potential genetic causes in couples with pure male idiopathic infertility by applying variant prioritization to whole exome sequencing. This result is very interesting in that it provides a new understanding of male infertility etiology including Mendelian causes of infertility. I think that further understanding of the genetic impact on infertility need to improve treatment options of infertility.

AU_we thank the Reviewer for his comments.

Reviewer #2: The authors exam the gene of idiopathic infertile men. Even though the treatment is same, it's a new field for idiopathic infertile men, and maybe further become routine work up. Several gene is related to cancer in POI women. The author also mention some of the gene is also related to cancer in idiopathic infertile men.

AU_we thank the Reviewer for her/his comments.

Reviewer #3: The author of the paper entitled "Whole exome data prioritization unveils the hidden weight of Mendelian causes of male infertility. A report from the first Italian cohort" analyzed the whole exome sequencing of the males with idiopathic infertility. I have some comments:

1. only 12 patients of the studied cases diagnosed to have pathogenic and likely pathogenic variants. Please discuss by details (case by case) how you are sure that their infertility is because of the identified variants and also discuss the defect of this method (NGS) to diagnose this type of idiopathic infertility.

AU_We thank the Reviewer for the suggestions that will improve the readability of the manuscript. In the revised version we include a thorough and detailed description of patient phenotype, the genetic variant and the consistency with the mutant gene reported as associated to that specific form of infertility.

Also, we discuss about the consistency of NGS data vs. Sanger sequencing by referring to our experience as Clinical Genomics ward in the hospital. At page 9 in the “Variants prioritization and classification” paragraph, we add “Since high coverage has been considered as a sufficient quality indicator [23], Sanger confirmation has not been performed. Our personal data based on internal procedure validation for more than 2000 genetic variants diagnosed through the NGS, first, followed by Sanger sequencing confirmation, revealed an excellent concordance of variant calling (PC, personal communication).”

2. Insert TESE outcome Column to Table 2 for all 12 patients and discuss the relationship of the identified variants to TESE outcome.

AU_ Table 2 now includes a new column describing TESE outcome for the 12 mutant patients. TESE results are discussed at page 15 of the manuscript.

---

## [Decision Letter · Decision Letter 1]

26 Jun 2023

Whole exome data prioritization unveils the hidden weight of Mendelian causes of male infertility. A report from the first Italian cohort.

PONE-D-23-10435R1

Dear Dr. Casari,

We’re pleased to inform you that your manuscript has been judged scientifically suitable for publication and will be formally accepted for publication once it meets all outstanding technical requirements.

Kind regards,

Nejat Mahdieh

Academic Editor

PLOS ONE

Additional Editor Comments (optional):

Reviewers' comments:

Reviewer's Responses to Questions

**Comments to the Author**

1. If the authors have adequately addressed your comments raised in a previous round of review and you feel that this manuscript is now acceptable for publication, you may indicate that here to bypass the “Comments to the Author” section, enter your conflict of interest statement in the “Confidential to Editor” section, and submit your "Accept" recommendation.

Reviewer #3: All comments have been addressed

2. Is the manuscript technically sound, and do the data support the conclusions?

Reviewer #3: Yes

3. Has the statistical analysis been performed appropriately and rigorously? 

Reviewer #3: Yes

4. Have the authors made all data underlying the findings in their manuscript fully available?

Reviewer #3: Yes

5. Is the manuscript presented in an intelligible fashion and written in standard English?

Reviewer #3: Yes

6. Review Comments to the Author

Reviewer #3: (No Response)

7. PLOS authors have the option to publish the peer review history of their article (what does this mean?). If published, this will include your full peer review and any attached files.

Reviewer #3: No

---

## [Editor Report · Acceptance letter]

27 Jul 2023

PONE-D-23-10435R1 

Whole exome data prioritization unveils the hidden weight of Mendelian causes of male infertility. A report from the first Italian cohort. 

Dear Dr. Casari:

I'm pleased to inform you that your manuscript has been deemed suitable for publication in PLOS ONE. Congratulations! Your manuscript is now with our production department. 

Kind regards, 

on behalf of

Dr. Nejat Mahdieh 

Academic Editor

PLOS ONE